# Soft End Effector Using Spring Roll Dielectric Elastomer Actuators

Hamish Lewis  and Min Pan *

Department of Mechanical Engineering, University of Bath, Bath BA2 7AY, UK; hl999@bath.ac.uk
* Correspondence: mp351@bath.ac.uk

**Abstract:** Dielectric elastomer actuators (DEAs) offer robust, high-energy-density solutions for soft robotics. The proposed end effector consists of three spring roll configuration DEAs, each acting as a robotic finger, using a 3M VHB-F9473PC adhesive membrane. Spring roll DEAs can be designed to achieve highly specialised actuations depending on the electrode patterning and structural supports. This allows a spring roll DEA-based soft end effector to be tailor-made by simply altering the electrode patterning. The lateral force, bending angle and response time of the actuator are measured experimentally and compared with the predictions of an analytical model. The cylindrical actuator measures 70 mm in length and 15 mm in diameter and achieves a lateral force of 30 mN, a bending angle of 6.8° and a response time of ≈1 s. Spring roll configuration DEAs are shown to reduce the effects of viscoelasticity seen in the membrane, making the actuator more controllable at higher voltages. The dielectric constant of the membrane is shown to be a limiting factor of actuation, with a decrease in dielectric constant resulting in larger actuation. The end effector successfully grips numerous light objects for extended periods, showing the applicability of spring roll DEAs for soft end effectors.

**Keywords:** dielectric elastomer actuator; soft end effector; multiple degrees of freedom; modelling; artificial muscle

## 1. Introduction

Dielectric elastomer actuators (DEAs) consist of an elastic dielectric membrane sandwiched between two compliant electrodes (Figure 1). DEAs have a very high energy density, allowing them to outperform other actuators when weight is constrained [1]. Actuating the bulk material allows for smaller form factors (subgram) and more resilient actuators due to their low inertia [2]. DEAs are promising because of their large strains, fast response, long lifetime, high efficiency and ability to be tailor-made by altering the electrode pattern [3–5]. As the electrode pattern dictates the actuation and is sprayed on the membrane using a stencil, it is easy for the end effector to achieve specialised actuations. Currently, one of the most popular dielectric elastomer (DE) materials used is the very-high-bond (VHB) polyacrylate film because of the large strains achievable and low cost. Another benefit to this material is that it can be pre-stretched, which offers substantial actuation performance, including higher efficiency and dielectric breakdown voltages, and reduces the probability of pull-in instability [6]. The thickness of the compliant electrode used also affects the performance of the actuator, with thinner electrodes generally being favoured over thicker ones provided that the electrode maintains its conductivity for the expected strains [7]. When a voltage is applied to the DEA, it becomes part of the circuit, i.e., a compliant capacitor. The physical changes that the membrane experiences once the voltage is applied can be modelled using circuit components [8].

Creating a dynamic model allows the performance of the actuator to be assessed across all scales, especially subgram actuators/robotics, where performance is shown to be highest when compared with other actuation methods [2]. DEAs can be modelled statically using an appropriate hyperelastic energy function and viscoelastic relaxation model [9]. The frequency response can also be modelled in a similar way to predict the amplitude

　　　　

of oscillation [10,11]. The dynamic response of a spring roll actuator during aperiodic actuation is not as well documented. As viscoelasticity causes a non-linear response in both the time and frequency domains [6], developing a complete model that can predict the behaviour of the actuator when gripping an object is essential to better understand the potential applications of spring roll DEAs.

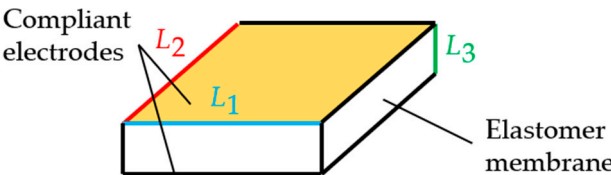

**Figure 1.** Working principle of DEA schematic.

When a voltage is applied to the electrodes, the membrane is compressed in the thickness direction due to the opposing stored charges, resulting in in-plane expansion and hence actuation [12]. Spring roll DEAs (shown in Figure 2a) consist of a compressed spring rolled in a dielectric elastomer (DE) membrane, with an electrode applied, and offer structural rigidity and some membrane pre-stretching. The response of the spring roll actuator strongly depends on the properties of the spring, with the bending angle $\theta$ decreasing with spring diameter $d_s$ and rate $K$. This study proposes a soft gripper using spring roll DEAs. The gripper consists of three spring roll DEAs attached to a polymer base, creating a soft gripper, as shown in Figure 2b.

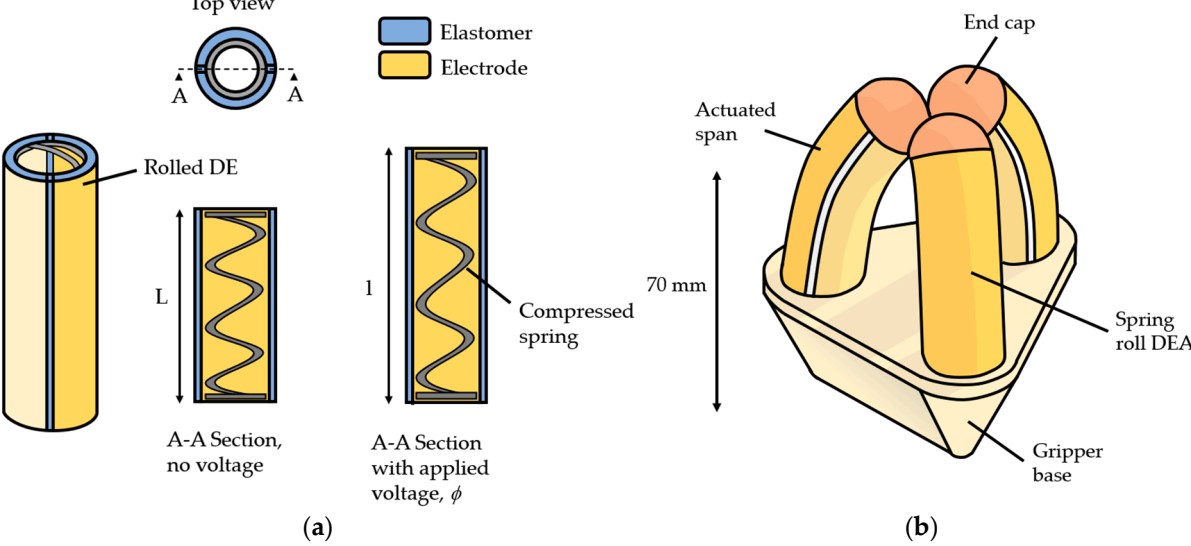

**Figure 2.** (**a**) A 2-DOF bending spring roll. For bending, only one span is actuated. For 1-DOF, both sides are actuated simultaneously resulting in a linear strain. (**b**) Spring roll DEA gripper concept schematic.

Voltages of 0.5 to 10 kV are typically seen to achieve large strains [9]. Elastomers are composed of long entangled polymer chains, bound by weak intermolecular bonds. DEs are similar except that, when placed in an electric field, the distribution of charges within the polymer chains is disrupted, causing the chains to become polarised [13]. The chains then acquire an electric dipole moment, resulting in a rotation due to the electric field (Figure 3a), with each pole being attracted to the oppositely charged electrode. Under large deformation, the polarisation of the chains is impeded (Figure 3b) [14]. The electromechanical response of the DEA can be uncoupled, i.e., investigated electrically and mechanically, and modelled computationally [9]. The DEs used are hyperplastic materials and behave highly non-

linearly due to the complex phenomena dictating the resulting in-plane expansion due to the conditions imposed upon the elastomer.

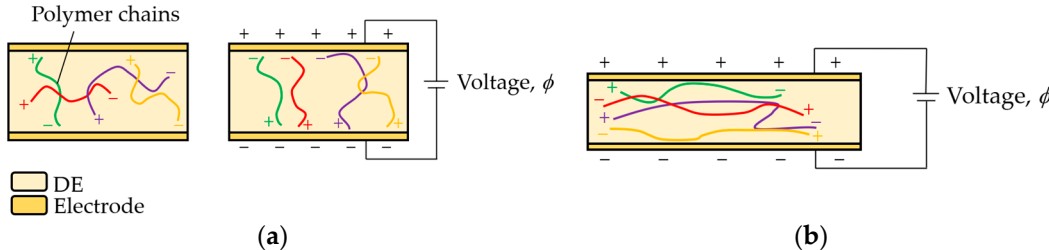

**Figure 3.** (**a**) Electric dipole moment, rotation of polymer chains in the direction of the applied voltage. (**b**) Electric dipole moment with deformation from applied voltage.

Numerous spring roll DEAs have been studied [6,15–18]; however, they are yet to be used for a soft gripper. Assessing the capability of a spring roll DEA-based end effector will contribute to improved designs and showcase their performance so that they can be considered for specialised applications. It also motivates new membrane materials to be developed, which have a direct impact on the performance of the actuator and will subsequently increase the impact of new innovations, which require lightweight actuators. DEAs may also be used as sensors, as, when compressed, a predictable voltage is produced. DEAs have been developed for applications such as soft end effectors, energy harvesting and lightweight crawling and flying robots [2,19,20]. An analytical model is first constructed and tested to predict the mechanical behaviour and response of the actuator. Actuator samples and a soft gripper are fabricated and compared with the analytical model to demonstrate the applicability of spring roll DEAs for use as soft end effectors.

## 2. Modelling and Experiment

Dynamic modelling is essential for the accurate analysis of the actuator behaviour. It allows parameters such as the spring rate, diameter and length to be chosen so that the response can be optimised for a given task. It also allows the effect of the material properties on the response to be analysed, which can improve the design of the actuator.

### 2.1. Equations of State for DEAs

To model the DEA, a 1D planar model is constructed and then used to predict the in-plane expansion and the curvature of the spring roll. The membrane has original dimensions $L_1$, $L_2$ and $L_3$. The membrane is subject to forces from the electric field generated by the voltage across the electrodes $\phi$ and mechanical forces $P_1$, $P_2$ and $P_3$, acting on the membrane, such as a pre-stretch, and these cause the original dimensions of the membrane to become $l_1$, $l_2$ and $l_3$, as shown in Figure 4. Each element of the actuated membrane is modelled, as shown in Figure 4b. The accumulated charge on each opposing electrode is represented by $Q$ and the resulting stretch of the membrane can be represented by the principal stretches, $\lambda_1 = l_1 / L_1$, $\lambda_2 = l_2 / L_2$, $\lambda_3 = l_3 / L_3$ [21]. The deformation of the membrane is shown by the curvature of the actuator and is assumed to be entropic [22].

The resulting work performed on the membrane (mechanical and charge from the applied voltage) can be represented using the Helmholtz free energy $F$, taken to be a function of four independent variables, $F(l_1, l_2, l_3, Q)$. Equation (1) is the free energy of the thermodynamic system $\Pi$ using the potential energy from the forces, compressed massless spring $K$ [6] and the applied voltage [12].

$$\Pi = F(l_1, l_2, l_3, Q) - P_1 l_1 - P_2 l_2 - P_3 l_3 - \phi Q + K L_1 (L_1 - l_1 \lambda_1) \tag{1}$$

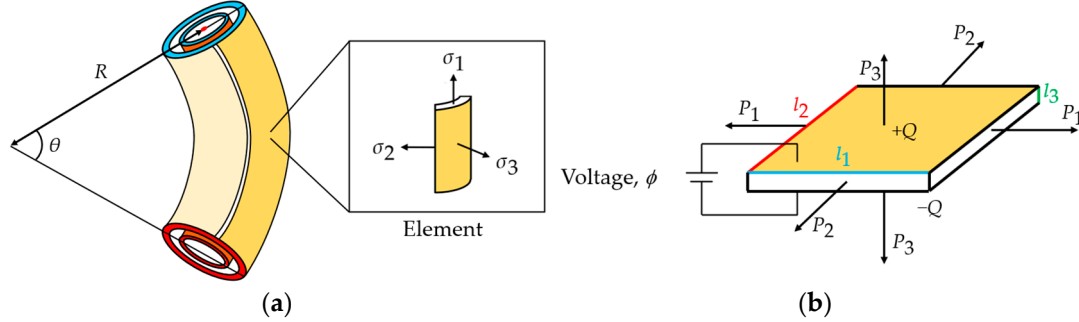

**Figure 4.** (**a**) Spring roll bending configuration with the right side actuated (shown in darker hue), resulting in a curvature of angle $\theta$ and radius $R$. An element of the actuated membrane is subjected to stresses when the voltage is applied, and for any forces acting on the membrane. Red surface represents the fixed end, blue surface represents the free end of the actuator and orange represents the massless spring. (**b**) Schematic of DE membrane element from (**a**).

This is a measure of the useful work of a closed system and states that energy is conserved; therefore, $F$ represents the maximum amount of work that the system can perform. Assuming that the forces and voltage are fixed, the free energy of the system, $\Pi(l_1, l_2, l_3, Q)$, can be defined as a function of four independent variables. Minimising this function represents an equilibrium state, i.e., the free energy has been minimised with respect to the variables. It can be assumed that the membrane undergoes homogeneous deformation; therefore, the nominal density of the Helmholtz free energy, $W = F/(L_1, L_2, L_3)$, can now be defined using the stresses, $\sigma_1\ P_1/(l_2l_3)$, $\sigma_2 = P_2/(l_1l_3)$, $\sigma_3 = P_3/(l_1l_2)$, the electric field, $E = \phi/l_3$, and the electric displacement, $D = Q/(l_1l_2)$. The volume of an elastomer undergoing large deformation can be assumed to be constant, as the change in the shape of an elastomer is typically much more significant. Equation (2) represents the constant volume and one of the equations of state of the membrane [22]:

$$\lambda_1\lambda_2\lambda_3 = 1 \tag{2}$$

Prior to the incompressibility assumption, each stretch was independent; however, only two are required now to fully represent the deformation, along with the electrical displacement, $D$. Setting $\lambda_3 = \lambda_1^{-1}\lambda_2^{-1}$ [9] and assuming that $W$ is a function of the three independent variables,

$$W = W(\lambda_1, \lambda_2, D) \tag{3}$$

The free energy of the system can be represented using Equations (1) and (3) [22]:

$$\begin{aligned}\Pi(\lambda_1, \lambda_2, D) =\ &L_1L_2L_3 \cdot W(\lambda_1, \lambda_2, D) - P_1L_1\lambda_1 - P_2L_2\lambda_2 \\ &- P_3L_2\lambda_1^{-1}\lambda_2^{-1} - \phi L_1L_2\lambda_1\lambda_2 D + K(L_1 - l_1\lambda_1) \cdot L_1\end{aligned} \tag{4}$$

Fixing the forces and voltage, an equilibrium state is achieved when Equation (4) is true. Setting the partial derivative of $\Pi(\lambda_1, \lambda_2, D)$ with respect to each independent variable = 0, i.e., $\partial\Pi(\lambda_1, \lambda_2, D)/\partial\lambda_1, \partial\lambda_2, \partial D = 0$, gives the other three equations of state [9]:

$$\sigma_1 - \sigma_3 = \lambda_1\frac{\partial W(\lambda_1, \lambda_2, D)}{\partial\lambda_1} - ED \tag{5}$$

$$\sigma_2 - \sigma_3 = \lambda_1\frac{\partial W(\lambda_1, \lambda_2, D)}{\partial\lambda_2} - ED \tag{6}$$

$$E = \frac{\partial W(\lambda_1, \lambda_2, D)}{\partial D} \tag{7}$$

The forces and voltage can now be determined given a suitable energy function, $W(\lambda_1, \lambda_2, D)$, for the incompressible DE [9]. Setting $D = \varepsilon E$, where $\varepsilon$ is the relative permittivity of the membrane, and integrating Equation (7) with respect to $D$ gives

$$W(\lambda_1,\ \lambda_2,\ D) = W_s(\lambda_1,\ \lambda_2) + \frac{D^2}{2\varepsilon} \tag{8}$$

This integration leaves two independent terms, the constant of integration, $W_s(\lambda_1, \lambda_2)$, which represents the Helmholtz free energy from the deformation of the elastomer, and the $D^2/2\varepsilon$ term, representing the Helmholtz free energy associated with the membrane polarisation. Note that $W_s(\lambda_1,\ \lambda_2)$ represents the free energy due to elastic stretching [22] or the strain energy density function used to model the deformation of the membrane. The electromechanical coupling is therefore a geometric effect given the expression $Q = L_1 L_2 \lambda_1 \lambda_2 D$. Equation (8) is known as the model of ideal dielectric elastomers [22] and can now be combined with the free energy of the system (Equation (4)) [8]:

$$\frac{\Pi(\lambda_1,\ \lambda_2)}{L_1 L_2 L_3} = W_s(\lambda_1,\ \lambda_2) - \frac{P_1}{L_2 L_3}\lambda_1 - \frac{P_2}{L_3 L_1}\lambda_2 - \frac{P_3}{L_1 L_2}\lambda_1^{-1}\lambda_2^{-1} - \frac{\varepsilon}{2}\left(\frac{\phi}{L_3}\right)^2 \cdot (\lambda_1\lambda_2)^2 + \frac{K(L_1 - l_1\lambda_1)}{L_2 L_3} \tag{9}$$

Note that the free energy of the system, $\Pi(\lambda_1,\ \lambda_2)$, is now a function of only two planar stretches as the voltage is fixed. Again, a state of equilibrium is reached at the minimum value of $\Pi(\lambda_1,\ \lambda_2)$, i.e., = 0. Equivalents to Equations (5) and (6) can now be written as [9]

$$\sigma_1 - \sigma_3 + \varepsilon E^2 = \lambda_1 \frac{\partial W_s(\lambda_1,\ \lambda_2)}{\partial \lambda_1} \tag{10}$$

$$\sigma_2 - \sigma_3 + \varepsilon E^2 = \lambda_2 \frac{\partial W_s(\lambda_1,\ \lambda_2)}{\partial \lambda_2} \tag{11}$$

where $\varepsilon E^2$ represents the Maxwell stress [23]. Acrylic membranes, especially the VHB series, suffer from viscoelastic effects. This results in long-term relaxations (hundreds of seconds [11]) and slower response times, which makes precisely modelling and controlling the actuator difficult. Additionally, any forces acting on the membrane, represented by $P_1$, $P_2$ and $P_3$ in Equation (9), will change the characteristics of the actuator and thus the response. Pre-stretching the film reduces the creep strain behaviour due to viscoelasticity and, therefore, may be an effective method to combat the difficulties with modelling and controlling the viscoelastic effects. The effect of the degree of pre-stretching on the performance of the actuator can be investigated [11].

The viscoelasticity was modelled using the non-linear viscoelastic DE model developed by Yang et al. [24] as it was developed using the same DE membrane. The model combines two parallel springs (one elastic and one inelastic with a viscous dashpot). Figure 5 shows the viscoelastic model composed of two parallel units. The upper unit consists of a spring $\alpha$ with shear modulus $M_\alpha$, and the lower unit consists of a spring $\beta$ with shear modulus $M_\beta$ and a dashpot with viscosity $\eta$. The spring $\beta$ and dashpot represent the non-linear time-dependent deviation from the equilibrium state, described by Equation (17). The deformations in spring $\alpha$, spring $\beta$ and the dashpot are characterised by $\lambda_1$ and $\lambda_2$; $\lambda_1^e$ and $\lambda_2^e$; and $\xi_1$ and $\xi_2$, respectively. The stretch of the parallel units is equal; therefore, $\lambda_1 = \lambda_1^e \xi_1$ and $\lambda_2 = \lambda_2^e \xi_2$. Limiting stretch parameters $J_{\lim,\alpha}$ and $J_{\lim,\beta}$ are used to represent the finite contour length of the membrane.

As the dashpot relaxes with time, the stored energy is dissipated, reducing the deformation capacity [11]. The strain energy density function $W_s$ can now be written as the sum of the contributions from the two units, $W_\alpha$ and $W_\beta$:

$$W_s = W_\alpha(\lambda_1,\ \lambda_2) + W_\beta(\xi_1\lambda_1,\ \xi_2\lambda_2) \tag{12}$$

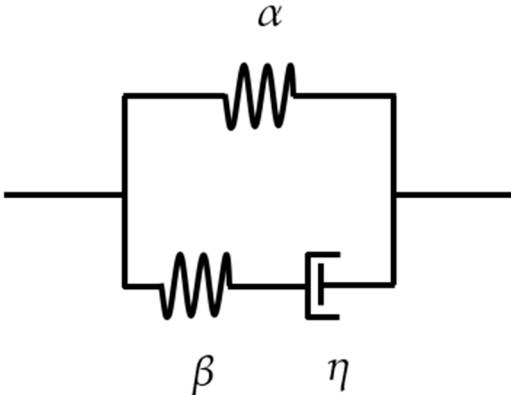

**Figure 5.** Standard linear solid viscoelastic model of dielectric elastomer using two parallel springs and a viscous dashpot.

The axial pre-stress of the film $\sigma_{1p}$ due to the compression of the spring may be represented by

$$\sigma_{1p} = W_\alpha(\lambda_{1p}) \tag{13}$$

The pre-stretch $\lambda_{1p}$ is a result of the spring's extension; $\lambda_{2p}$ is assumed = 1 as the spring restricts any lateral expansion of the membrane. A lower value for $K$ increases the bending angle; therefore, as $K$ is low, the pre-stress from the spring can be assumed = 0 [6]. Maintaining the incompressibility assumption sets $L_3 = L_3/\lambda_1$. The non-linearities in the deformation of the film are now represented and the deformation may be predicted and compared with the experimental results. Therefore, setting $\sigma_2 = \sigma_3 = 0$ [25] and $\lambda_2 = 1$ gives

$$\varepsilon\left(\frac{\phi\lambda_1}{L_3}\right)^2 = W_\alpha(\lambda_1,\ \lambda_2) + W_\beta(\xi_1\lambda_1,\ \xi_2\lambda_2) - \frac{K(L_1 - l_1\lambda_1)}{L_2 L_3} \tag{14}$$

Written in full using the Gent energy function [26],

$$\varepsilon\left(\frac{\phi\lambda_1}{L_3}\right)^2 = \frac{M_\alpha\left(\lambda_1^2 - \lambda_1^{-2}\right)}{1 - \left(\lambda_1^2 + \lambda_1^{-2} - 2\right)/J_{\lim,\alpha}} + \frac{M_\beta\left(\lambda_1^2\xi_1^{-2} - \xi_1^4\lambda_1^{-2}\right)}{1 - \left(\lambda_1^2\xi_1^{-2} + \xi_1^4\lambda_1^{-2} - 2\right)/J_{\lim,\beta}} - \frac{K(L_1 - l_1\lambda_1)}{L_2 L_3} \tag{15}$$

where $J_{\lim,\alpha}$ and $J_{\lim,\beta}$ are material constants and represent the limiting stretches of spring $\alpha$ and spring $\beta$, respectively. Only the membrane regions with electrodes will be subjected to the Maxwell stress. The viscoelastic relaxation time $t_v$ is set $= \eta/M_\beta = 50$ s [11]. Equation (15) can be solved for $\lambda_1$ as a function of time $t$, allowing the dynamic behaviour of the actuator to be assessed. Model-specific material parameters are given in Table 1.

**Table 1.** Gent energy function model parameters for 3M VHB-F9473PC adhesive membrane [24].

| Material Parameter | Value |
| --- | --- |
| $J_{\lim,\alpha}$ | 115 |
| $J_{\lim,\beta}$ | 70 |
| $M_\alpha$ | 16,000 |
| $M_\beta$ | 45,000 |
| $\eta$ | $2.25 \times 10^6$ |

The bending angle of the 2-DOF roll, $\theta_{2-\text{DOF}}$, is assumed to have constant curvature and estimated using [18]

$$\theta_{2-\text{DOF}} = \frac{2S}{\pi r} \tag{16}$$

where $S$ is the stroke of the actuator and $r$ is the radius of the spring. The viscoelastic relaxation of the membrane can be represented by the deformation rate $\frac{d\xi_1}{dt}$ and viscosity $\eta$ of the dashpot, shown using the Gent model [24]:

$$\frac{d\xi_1}{dt} = \frac{1}{3\eta}\left[\frac{M_\beta\left(\lambda_1^2\xi_1^{-2} - \xi_1^4\lambda_1^{-2}\right)}{1 - \left(\lambda_1^2\xi_1^{-2} + \xi_1^4\lambda_1^{-2} - 2\right)/J_{\lim,\beta}}\right] \tag{17}$$

The axial stress from the compressed spring when the DEA is released after rolling is calculated using Equations (15) and (17).

### 2.2. Charge Control and Leakage Current

The membrane cannot be assumed to be a perfect insulator, and therefore part of the charge that accumulates on the electrodes, $Q_C$, leaks through, $Q_{\text{leak}}$, as shown in Figure 6. This leakage current consists of electronic and ionic conduction, where charged particles within the material complete the circuit due to the applied field [8,27].

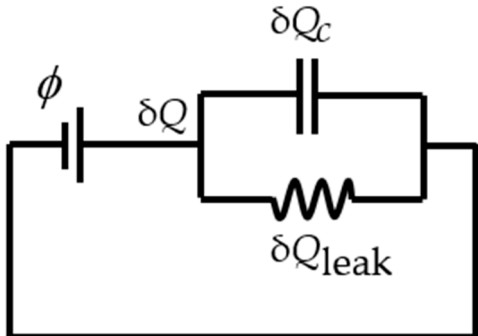

**Figure 6.** Leakage current model.

The process of charging and discharging the capacitor is not instant and can be represented by

$$\delta Q = \delta Q_C + \delta Q_{\text{leak}} \tag{18}$$

where $\delta Q$ is the charge moving in the wire. Dividing Equation (18) by $dt$ gives the change in current with time:

$$i = \frac{\delta Q_C}{dt} + i_{\text{leak}} \tag{19}$$

where $i = dQ/dt$ represents the current in the wire, $\delta Q_C/dt$ represents the rate of change in the charge on the electrodes and $i_{\text{leak}} = dQ_{\text{leak}}/dt$ represents the leakage current through the membrane [28]. The capacitor and resistor circuit transfer function is then generated and used to create the first-order lag response of the capacitor charging with the leakage current. Equation (20) shows the resulting transfer function:

$$\frac{\phi_C}{\phi} = \frac{R_{\text{leak}}}{s \cdot R \cdot R_{\text{leak}} \cdot C + R + R_{\text{leak}}} \tag{20}$$

where $\phi_C$ is the voltage across the capacitor, $R$ is the resistance used to limit the current from the power supply, $R_{\text{leak}}$ is the leakage current equivalent resistance, $C$ is the capacitance of the DEA and $s$ is the complex parameter from the Laplace transform. The charge on the capacitor is then converted to an equivalent field, where the leakage current is modelled using a resistor, $R_{\text{leak}}$ [27,28]:

$$R_{\text{leak}} = \frac{L_3}{l_1 l_2 c_0 e^{\left(\frac{\phi}{\phi_B}\right)}} \tag{21}$$

where $\phi_B$ is the breakdown voltage of the membrane, and the membrane conductivity $c_0 = 2.159 \times 10^{-14}$ [27]. The discharging process is equivalent and inverse.

## 3. Method

### 3.1. Fabrication of Spring Rolls

The membrane used is 3M VHB F9473PC ($L_3 = 250$ μm) and is cut into sheets and sprayed with Kontakt Chemie Graphit 33 Conductive Lacquer, an electrical conductive coating acting as the electrode, using an MDF stencil. Copper foil tape is added so that the voltage is applied to alternating electrodes, as seen in Figure 7a. The spring is first compressed, and then the membrane is rolled around the spring. To ensure that no air is trapped between the layers, the membrane is pressed against the spring as it is rolled. The spring roll DEA is then released and left to relax until the spring and membrane are in equilibrium. The axial pre-stress of the membrane can be determined using the length of the relaxed spring. Multiple samples are fabricated and tested to ensure the reliability of the fabrication process.

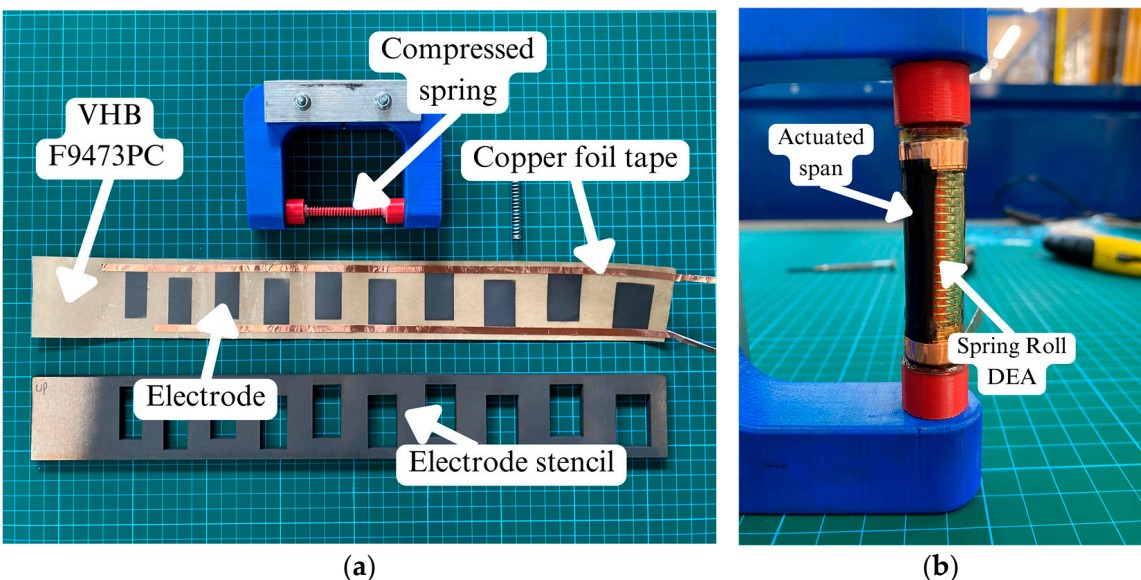

**Figure 7.** (**a**) Fabrication process components; (**b**) completed spring roll actuator.

This process can be automated using an electric motor and jig; however, as the membrane does not require constant tension when rolling, as is required when pre-stretching, this is not necessary for coherent rolling.

### 3.2. Experimental Setup

The actuator is fixed at the base, placed in front of a grid of known size and actuated in increments of 1 kV from 0 to 6 kV (Figure 8b). The response is captured using a camera and then calibrated and analysed using the motion capture software Kinovea. To measure the lateral force, the actuator is fixed horizontally and allowed to relax. The end cap is then fastened to a cantilever beam and strain gauge and actuated in increments of 1 kV from 0 to 6 kV. Multiple measurements for each voltage increment are taken. The bending angle and lateral force are measured multiple times per second whilst the voltage is applied to generate time series data, allowing the dynamic response to be analysed and assessed across the entire stroke of the actuator. Standard insulated wires cannot be used to connect the actuators to the power supply as their inertia affects the response of the actuator.

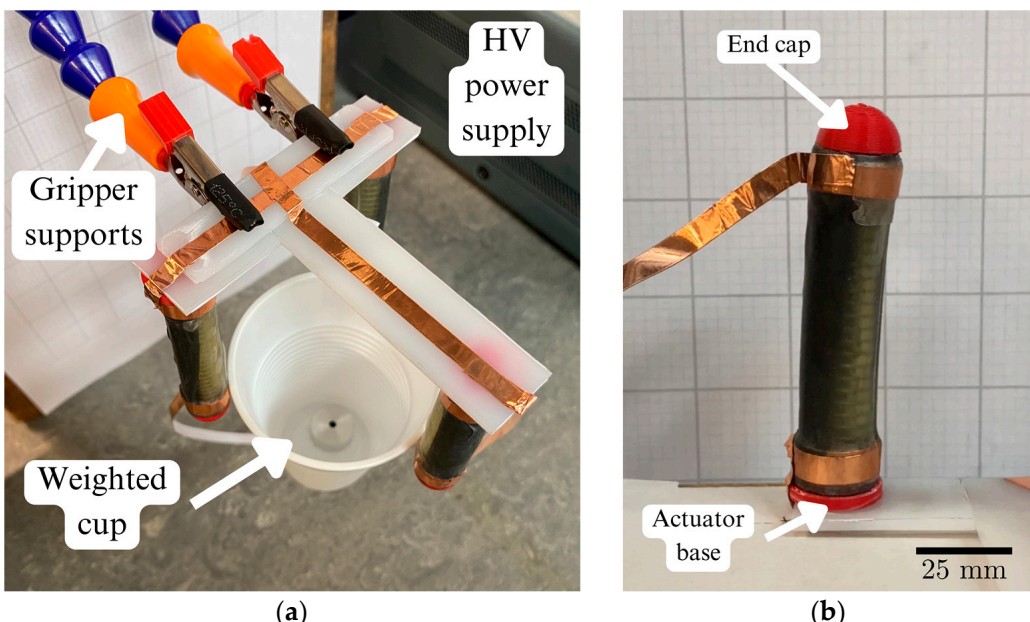

**Figure 8.** (**a**) Gripper holding a 35 g mass in a cup. The gripper consists of three single actuators arranged radially. (**b**) Single actuator bending test, shown in an actuated state. A lateral deflection of ~5 mm was achieved for the actuator design used.

The gripper was fabricated using polymer sheets and copper foil tape, as shown in Figure 8a. A membrane layer was placed over the foil tape at the effecting end of the actuator. Gaps between the edges of the membrane and electrodes were left to ensure that no arcing occurred between or within the membrane layers. The electrode regions were sized and spaced to maintain alignment when rolled, increasing with the roll diameter.

## 4. Results

The total mass of the gripper was 36 g and it was able to hold objects with masses of up to 35 g, as shown in Figure 8a. The fingers were angled perpendicularly to the base of the gripper and were arranged radially with a radius of 35 mm. The fingers were directed towards the centre, with the actuated spans facing outwards such that the gripper closed when actuated. To test the performance of the gripper, the cup was located within the grasp of the gripper, and the gripper was then actuated and lifted. When the voltage was removed, the cup fell after ~2 s because of the viscoelastic effects of the membrane. The gripper showed repeatable actuation with a predictable response and no preference regarding the direction of initial actuation was observed.

### 4.1. Static Response

The voltage-controlled actuation is shown in Figure 9. The bending angle and lateral force increased quadratically with the voltage and fit with the predictions of the model. The voltage was applied for $t = 10$ s before measurements were taken to assess the dynamic elastic and viscoelastic predictions of the Simulink model. Small deviations were seen from the trend line across all experimental data points. The bending angle at 5 to 6 kV was lower than the model predictions and was likely due to defects in the membrane, imperfections introduced during the fabrication process and thickness variations in the regions of the membrane patterned with compliant electrodes.

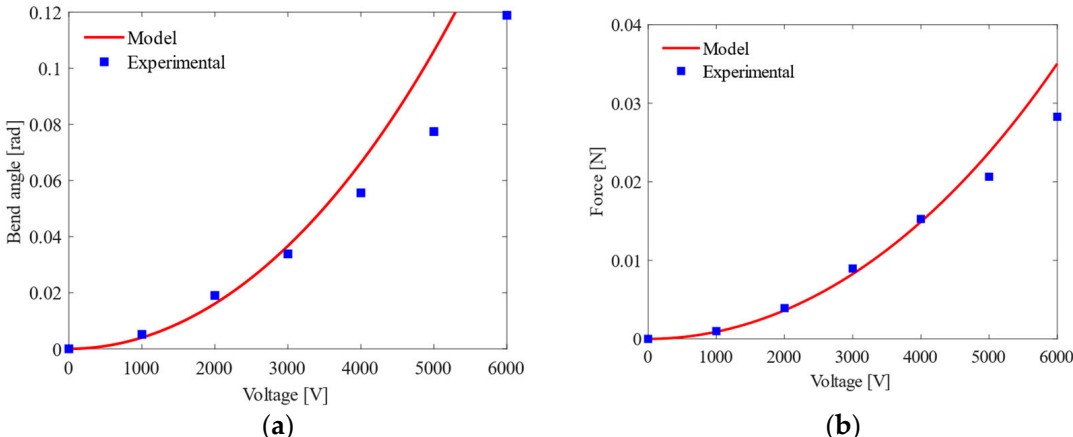

**Figure 9.** (**a**) Voltage-controlled bend angle for $t = 10$ s or $t = t_v/5$; (**b**) maximum lateral force generated for $t = 10$ s, with a resolution of $\pm 0.005$ N.

Dielectric breakdown occurs at the breakdown voltage $\phi_B$ and below, given sufficient time for the viscoelastic effects to reduce the membrane thickness according to the relationship

$$\phi_B = \frac{EL_3}{\lambda_1} \tag{22}$$

where $E$ is the dielectric strength. When actuated, the gripper is more vulnerable to external forces, causing dielectric breakdown as a slight decrease in the membrane thickness is able to reduce $\phi_B$ below the supplied voltage.

The model accurately predicted the lateral force of the actuator at lower voltages, becoming less accurate at higher voltages due to membrane defects. Another likely cause is that only forces with components normal to the beam are represented correctly by the experimental data. As the actuator has a constant curvature, there will be some component of force not acting normal to the beam, which are not measured. This effect becomes more apparent at 4 kV and above (Figure 9).

*4.2. Dielectric Properties of Membrane*

From Figure 10, a lower value of the dielectric constant results in a larger breakdown voltage and thus allows for a much larger stretch before breakdown occurs. This is due to the breakdown strength of the material being inversely proportional to the dielectric constant and agrees with the literature [29,30]. The increase in $\lambda_{max}$ from $k = 4$ to $k = 2$ is significant and would result in an increase in bend angle proportional to the additional stretch. The theoretical maximum was plotted, $k = 1$, resulting in a much larger actuation. The lowest dielectric constant of polymers is usually around 2 [30], which would suggest that the membrane chosen is not optimal in terms of dielectric constant, and the performance of the DEA would see a significant increase upon reducing the dielectric constant, assuming that the material properties are similar.

The breakdown line for each voltage is not at a constant voltage as the thickness of the membrane reduces with actuation, and therefore the maximum voltage that the membrane can withstand before breakdown decreases with $\lambda_1$ as shown.

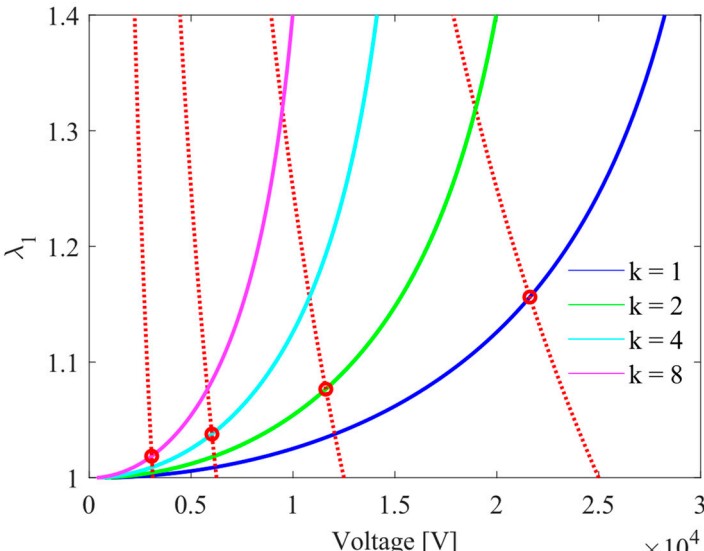

**Figure 10.** Membrane actuation for varying dielectric constant, *k*. Breakdown line for each value shown by red dashed line, with the intersection point (maximum stretch of the membrane, $\lambda_{\max}$) shown by red circles for each dielectric constant. $k = 3$ for VHB-F9473PC.

### 4.3. Dynamic Response

A predictable dynamic response is observed, as shown in Figure 11a, with the viscoelasticity effects being represented by the configuration shown in Figure 5. A voltage of 3 kV was used as this was the highest voltage accurately predicted by the model. The discrepancy between the magnitude of the responses is likely due to efficiency losses in the membrane from defects, air bubbles, misaligned electrode spans and non-actuated electrode regions near the ends of the actuator, visible in Figure 7b. A clear experimental response curve is seen; however, the material constants used to model the membrane (seen in Table 1) are likely inaccurate as the response predicted by the model does not align with the observed response of the actuator. The initial sharp increase seen in Figure 11a is also dictated by the current from the power supply and can be adjusted to alter the power of the actuator, with a larger current increasing the speed of the response. The response time can be reduced by decreasing the viscosity and/or the dielectric constant of the membrane, and it would also possible through the selection of a different DE membrane with advancements in membrane material science.

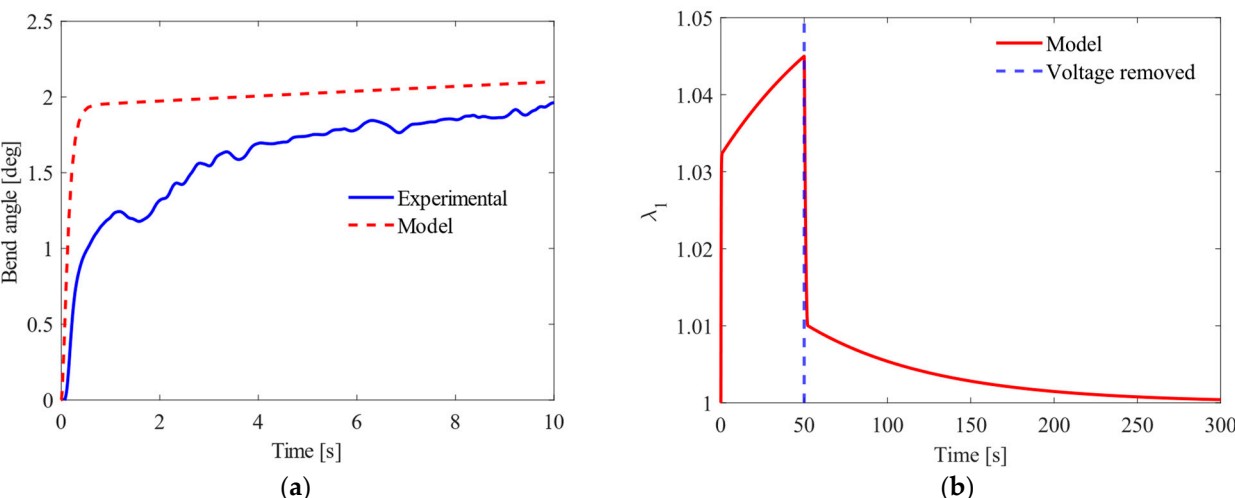

**Figure 11.** (**a**) Effect of spring on controllability of actuator for 3 kV; (**b**) viscoelastic relaxation time of actuator when actuated for 50 s at 5 kV.

Figure 11b shows the viscoelastic relaxation time after actuation is held for 50 s. The actuator cannot hold a 6 kV voltage for 50 s as dielectric breakdown will occur due to the membrane thickness reducing. Once the voltage is removed, the time taken to reach $\lambda_1 = 1$ is roughly five times longer than the initial actuation period (left of the blue dashed line). This limits the period of actuation for continuous operation required for good performance and controllability.

## 5. Discussion

The actuators display a predictable and reliable response for short-term actuations, where the dashpot is not fully extended and is responsible for a small portion of the total actuation. Should the voltage be applied for longer periods, as shown in Figure 11b, the membrane must be allowed to relax before accurate actuation can be performed again. The controllability of the actuator reduces significantly for multiple long-term actuations because of this. Incorporating the spring into the actuator design improves the controllability by forcing the membrane to adopt its shape, allowing the bend angle to be calculated easily with a constant curvature assumption. This also limits the stretch seen in the membrane once the spring is fully extended, allowing for actuations to be held indefinitely with a fixed and predictable relaxation period. The spring increases the relaxation period as the spring is re-compressed, decreasing the shear modulus of the upper unit of the viscoelastic model, $M_\alpha$. The spring can be replaced with any elastic material, allowing the actuator to be scaled down. As the electrode pattern on the membrane allows for multiple degrees of freedom, on subgram scales, it is possible for a single membrane to carry out multiple functions with an appropriate supporting structure and electrode patterning. Specifically, 3D-printed compliant mechanisms are likely good candidates for use as supporting structures as they can constrain actuation, mitigating the viscoelastic effects of the membrane.

Each sample tested gave repeatable results, which demonstrates the reliability of the fabrication process and experimental setup. Samples where the dielectric breakdown occurred were no longer able to actuate. Samples that were punctured or damaged often displayed arcing. Given the voltage used for actuation, and the large membrane area required for actuation, the current design requires altering to improve the safety of the actuator such that punctures or defects do not result in arcing or damage to the surrounding environment. Samples showed resilience to wear, dropping, bending to $\theta = 180°$ and tight gripping, which caused no visible damage or detectable performance losses. Soldering directly to the copper foil tape did not damage the membrane.

The Gent model values, specifically $M_\alpha$, $M_\beta$ and $\eta$, are likely inaccurate because they are generated from a similar but different membrane (VHB 4910). The main difference between the membranes is the thickness, with VHB 4910 being 1 mm thick, as opposed to 0.25 mm. Material tests must be carried out on VHB F9473PC for the values to be accurate. The effects of the pre-stretch on the membrane were assumed to be negligible as a ratio of 1.03 was applied. Increasing the number of rolls was not investigated. The performance of the actuator is compared with previous works in Table 2.

**Table 2.** Comparison of actuator performance with previous works.

| Parameter | 2-DOF Roll (Version 1), 2004 [18] | 2-DOF Roll, 2016 [16] | This Work |
|---|---|---|---|
| Actuator mass | 29 g | - | 10 g |
| Actuator length | 68 mm | 40 mm | 70 mm |
| Dielectric elastomer (pre-stretch ratio) | VHB-4910 | VHB-4910 (3, 5) | VHB-F9473PC (1.03, 1) |
| Maximum operating voltage | 5.5 kV | 5 kV | 6 kV |
| Maximum stroke | - | 8.4 mm | - |
| Maximum bending angle | 60° | 75.3° | 6.8° |
| Maximum force | 1.68 N | 0.7 N | 0.03 N |
| No. of rolls | 20 | 14 | 10 |
| Dashpot fully extended | Yes | Yes | No |

The values obtained from previous works represent the actuation achieved once the dashpot has been fully extended. However, this can take hundreds of seconds, as shown in Figure 11b, and only partially represents the performance of the gripper. The actuation expected is, therefore, lower if the dashpot is not allowed to fully extend.

The non-pre-stretched membrane does not perform as well as the pre-stretched membrane; however, some supporting structures are not capable of maintaining large pre-stretch ratios due to either the large relative stress required to maintain a large pre-stretch or due to geometrical constraints and thus cannot make use of the improved performance.

## 6. Conclusions

DEAs are highly interdisciplinary and introduce challenges that demand further scientific exploration. DEAs that use springs or other compliant structural supports can be manufactured for purpose inexpensively and provide actuation and locomotion solutions for lightweight robotics. Advancements in membrane material science will lead to the development of new DE membranes, offering performance increases. Decreasing the dielectric constant or the viscosity of the DE membrane will increase the performance of DEAs and allow more applications to benefit from their use. Commercial success depends on new innovations requiring inexpensive, lightweight actuation solutions.

The actuation of a helical spring-based DEA was modelled and compared with experimental data. The analytical model predicted the behaviour of the actuator, with discrepancies identified. Methods of mitigating the viscoelastic effects of the membrane were identified. The dielectric constant and the breakdown strength of the membrane limited the performance of the actuator. The model may be used to identify materials and assess actuator designs with desired parameters. Spring roll DEAs can be used for collision-resistant, lightweight grippers with good controllability. The spring can be substituted for a lightweight structural support, allowing the gripper to be easily adapted for the application. Multiple regions of a single membrane can be supported and patterned with electrodes for specific tasks, including locomotion or for use as a soft gripper, especially for subgram robots. Future designs can make use of a membrane with a lower dielectric constant and viscosity and 3D-printed compliant structures to improve actuation.

**Author Contributions:** Conceptualisation, H.L. and M.P.; methodology, H.L. and M.P.; formal analysis, H.L.; investigation, H.L.; data curation, H.L.; writing—original draft preparation, H.L.; writing—review and editing, M.P.; supervision, M.P.; funding acquisition, M.P. All authors have read and agreed to the published version of the manuscript.

**Funding:** This research was funded by the University of Bath Alumni Fund, grant number F1920A-RS02.

**Data Availability Statement:** Not applicable.

**Acknowledgments:** The authors thank Runan Zhang for supporting the study.

**Conflicts of Interest:** The authors declare no conflict of interest.

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
