# Peer review of "Soft End Effector Using Spring Roll Dielectric Elastomer Actuators"

_actuators, doi:10.3390/act12110412_

Round 1

Reviewer 1 Report

Comments and Suggestions for Authors

This work proposes an end effector using spring roll dielectric elastomer actuators. The authors built an analytical model that includes a standard linear solid viscoelastic model of dielectric elastomer to predict the dynamic response of the actuators. While there were some discrepancies between experimental and theoretical results, the authors were able to identify the possible sources. The one concern that the reviewer needs to express is that the motivation for this research is not well explained in the abstract and the introduction. For example, that spring roll DEAs “are yet to be used for a soft gripper” (page 3, line 72) does not justify the necessity to carry out this work. The authors are therefore advised to revise the manuscript before it could be accepted for publication.

Author Response

Thank you for taking the time to review this manuscript.

Reviewer 2 Report

Comments and Suggestions for Authors

1.     The authors need to show the advantages and performance of their rolled DEA in comparison with literature precedents with similar structures.

2.     How was the axial stress calculated from the length of the relaxed spring?

3.     Is there a preferred direction for the spring-roll DEA to bend upon its first actuation?

4.     The response time (~2 s) of the spring-roll DEA seemed to be affected by the viscoelasticity of the acrylate elastomer. Is there any way to reduce the response time (as well as hysteresis)?

5.     How was the lateral force measured? What force gauge did the authors use, and what is the resolution?

6.     Have the authors tried to increase the number of rolls to increase the output?

Comments on the Quality of English Language

The writing is good.

Author Response

(The authors gave the same response as above.)

Reviewer 3 Report

Comments and Suggestions for Authors

Please check the attached file

Author Response

(The authors gave the same response as above.)
